https://doi.org/10.1038/s43856-022-00164-x | OPEN
# Dental biorhythm is associated with adolescent weight gain

Patrick Mahoney [1✉], Gina McFarlane[1], Carolina Loch[2], Sophie White[2], Bruce Floyd[3], Erin C. Dunn[4], Rosie Pitfield [1], Alessia Nava [1] & Debbie Guatelli-Steinberg[1,5]

## Abstract

**Background** Evidence of a long-period biological rhythm present in mammalian hard tissue relates to species average body mass. Studies have just begun to investigate the role of this biorhythm in human physiology.

**Methods** The biorhythm is calculated from naturally exfoliated primary molars for 61 adolescents. We determine if the timing relates to longitudinal measures of their weight, height, lower leg length and body mass collected over 14 months between September 2019 to October 2020. We use univariate and multivariate statistical analyses to isolate and identify relationships with the biorhythm.

**Results** Participants with a faster biorhythm typically weigh less each month and gain significantly less weight and mass over 14-months, relative to those with a slower biorhythm. The biorhythm relates to sex differences in weight gain.

**Conclusions** We identify a previously unknown factor that associates with the rapid change in body size that accompanies human adolescence. Our findings provide a basis from which to explore novel relationships between the biorhythm and weight-related health risks.

## Plain language summary

The human body undergoes cyclic changes such as the daily cycle of sleeping and waking, and monthly menstruation. This study calculated one cycle that can be tracked through the growth of children's milk teeth. The timing of the cycle in different children was compared to changes in body size that occurred when these children were in puberty. A link was seen between the children's cycle and the weight they gained over 14-months. Adolescents with a faster cycle typically weighed less each month and gained less weight over 14 months compared to those with a slower cycle.

[1] School of Anthropology and Conservation, University of Kent, Canterbury, UK. [2] Sir John Walsh Research Institute, Faculty of Dentistry, University of Otago, Dunedin, New Zealand. [3] School of Social Sciences, University of Auckland, Auckland, New Zealand. [4] Department of Psychiatry, Harvard Medical School and the Massachusetts General Hospital, Boston, MA, USA. [5] Department of Anthropology, The Ohio State University, Columbus, OH, USA. ✉email: p.mahoney@kent.ac.uk

Human adolescence is a period of rapid change in body size following the onset of puberty[1]. Sex-specific increases in lean muscle, bone mass, stature, and the amount and distribution of subcutaneous and total body fat[2–4] contribute to extensive gains in body size[2,5–8]. These shifts vary by the stage of puberty for males and females[9,10]. Adolescents can gain 8.3–9.0 kg a year[2,6] depending upon genetic[11–14] and environmental factors such as dietary habits[6] and activity levels[15,16].

The hypothalamus plays a pivotal role in the pubertal transition. It is a region of the brain that stimulates the release of hormones and regulates food intake and energy expenditure. Under the influence of growth hormone and insulin-like growth factor-I in early adolescence, the steroid hormone oestradiol creates the main growth spurt responsible for body size changes in both sexes (testosterone is converted in males)[17,18]. The change in body size is mediated via the hypothalamic-pituitary-gonadal axis[17,18].

Life on earth is regulated by biological rhythms. Some are daily rhythms linked to the light-related circadian cycle[19,20]. Others are longer than 24-h with an infradian cycle. Evidence of the infradian cycle is present in a range of organisms (such as tree rings) and mammalian physiological systems[20–23]. For humans, a near 7-day rhythm has been identified in adult heart rate, core body temperature, excretion of metabolites and salt and blood pressure during pregnancy[24–29].

Accumulating evidence suggests an infradian biorhythm may act upon the mammalian hypothalamus to regulate cell growth and body mass[30,31]. Microscopic-layered structures of mammalian teeth retain evidence of this rhythm. In human tooth enamel, the rhythm is referred to as Retzius periodicity (RP)[32] (Fig. 1). RP forms through a circadian-like process, occurring with a repeat interval that can be measured through histology with a resolution of days. The rhythm is consistent within the permanent molars of individuals[33,34] that do not retain evidence of developmental stress[35]. RP relates to the period in which tooth enamel forms. For human primary molars, this is the two-year period following birth[36]. The human modal RP has a near 7-day cycle[33,34,37,38] but varies from five to 12 days[38,39] when compared between individuals. Higher RP values occurring over more days suggest a slow underlying biorhythm. Lower RP values suggest a faster biorhythm.

Researchers during the 1990s[40,41] suggested variation in RP might relate to species-specific average body mass. Interspecific studies (meaning studies comparing different species) confirmed these observations revealing that, with exceptions[42,43], RP-biorhythm was higher ('slower') in larger-bodied living species, including anthropoids[30,31,44–46]. In these studies, biological pathways connecting RP and interspecific variation in body size were proposed. Larger-bodied species were suggested to attain their greater adult size through a slower biorhythm that produces slow growth rates over long periods of time, relative to the faster biorhythm of smaller-bodied species[30,31]. This pathway has emerged as a key hypothesis for advancing understanding of the evolution of primate life history[31].

Interspecific relationships are not always found within species[47]. But when the underlying cause is similar across different taxonomic levels, then similar biological relationships can be present within and between species[48]. The hypothalamus has a central role in human growth[17,18]. Our previous studies suggest aspects of human growth may relate to RP-biorhythm. Specifically, we have shown that the size of microscopic canals that house blood vessels in human adolescent ribs relates to RP[49]. Larger canals facilitate greater blood flow and nutrient transfer[50,51]. We observed higher RP values correspond with increased deposition of primary bone in humeri of young children[52]. These studies hint at a biorhythm underlying RP that influences rates of cell proliferation during the childhood growth years.

Studies of adult humans indicate taller adults tend to have lower RP values compared to shorter individuals[53–55]. The biorhythm appears to have a limited association with adult human weight[55]. Researchers utilised the height data from adults to hypothesise a biological pathway for human growth that differs from the interspecific pathway[30,31]. As the duration of human growth is constrained, relative to interspecific variation in growth periods, the biorhythm might accelerate to increase cell proliferation to achieve greater body size[30]. Thus, in contrast to the interspecific positive correlation between the duration of growth periods and body size and RP, the idea is that stature and RP should correlate negatively in humans. Currently, however, evidence of the biorhythm in relation to human growth[49,52] is limited.

Here, we calculate the biorhythm from primary molars in relation to weight gain for 61 children (average starting age = 10.33 years) from Dunedin, southern New Zealand, over a period of 14 months between September 2019 to October 2020. Adolescent weight is of particular interest because of the substantial gains during puberty that are driven by the hypothalamus. We demonstrate that adolescents with a faster biorhythm gain less weight over 14 months and have the smallest change in their body mass index (BMI) compared to adolescents with a slower biorhythm.

## Methods

**Participants, dental samples, study design and ethics**. The 61 participants ($n = 34$ females and $n = 27$ males) were selected from a larger cohort that were part of the Biorhythm of Childhood Growth project. The BCG is an ongoing prospective cohort study that investigates childhood development in middle-income children from Southern New Zealand[56]. Participants attended primary schools at the start of the project and then intermediate schools (see acknowledgements) within Dunedin city, New Zealand. 49 participants were of New Zealand European ethnicity. Six participants were of mixed heritage, either New Zealand European/Māori or New Zealand European/Pasifika. Six participants were either Māori, Pasifika, Iranian or of mixed Swiss/Korean heritage.

Naturally exfoliated primary molars were collected from all BCG participants ($n = 125$ children) and $n = 61$ were randomly selected from these for the current analyses based upon histology criteria (see methods). RP was calculated for each participant directly from their naturally exfoliated primary molars, which was compared to measures of that individual's weight and BMI. RP was calculated by one of us, GM, in the United Kingdom independently and blind to the weight and height data recorded in New Zealand by another author (SW).

We focused on primary molars only, as RP is a sequence for some individuals that can change between tooth types along the tooth row[33]. All deciduous molars, both maxillary and mandibular, were naturally exfoliated during the project. They were collected once a month during the monthly measurement of the growth variables. Molars with accentuated markings (also known as stress lines) were excluded as RP can sometimes change on either side of a stress marking[35].

Additional measures were incorporated into our study design so we could identify their effect on potential relationships between RP and weight gain. Adolescence typically commences in females (age 9 to 12 years) before males (age 11 to 14 years)[1,2]. Peak growth in height is greater for males but occurs sooner for females[57]. Because of these sex differences in the timing of adolescence, we expected females to gain more weight and height than males over the course of 14 months. If the biorhythm relates to adolescent weight/BMI gains, then there should be sex differences in these relationships.

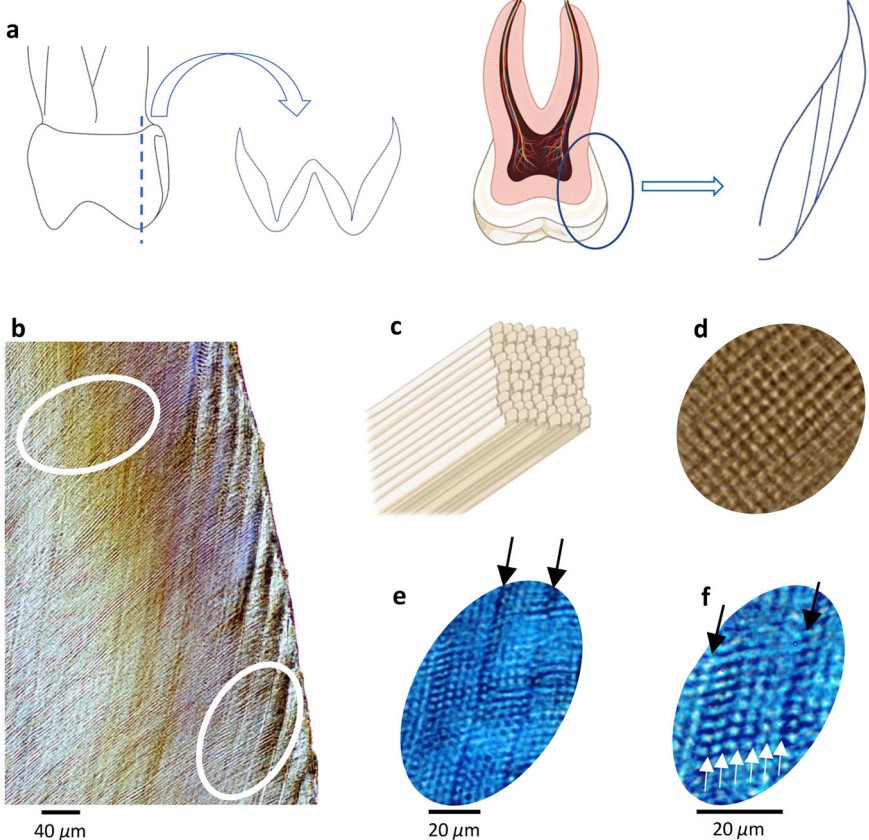

**Fig. 1 Calculating RP-biorhythm in human primary molars. a** Sectioned primary (deciduous, 'milk') molar. An arrow pointing to lateral enamel with Retzius lines on the far right. **b** Thin section through enamel with Retzus lines to the right (lower white circle). The Upper white circle overlays tubular enamel rods, which formed as groups of cells (named ameloblasts) lay down new enamel as a tooth crown develops. **c** A record of ameloblast pathways are preserved in teeth as enamel rods. **d** Daily cross striations. Enamel deposition by ameloblasts is interrupted every 24-h producing regions along rods that have relatively less mineral. When prepared and examined under a microscope, these differences in mineralisation along rods appear as cross striations. This occurs because variation in mineralisation alters the refractive index of light transmitted by a microscope, producing the striations. Cross striations are used to calculate Retzius periodicity. **e** Black arrows point to Retzius lines in primary molar enamel. **f** White arrows point to cross striations and 6 days of enamel formation between two adjacent Retzius lines giving a Retzius periodicity of 6 days. Parts of Fig. 1 (part of the panel **a** and all of panel **c**) were created using a template from BioRender.com (2022).

Many factors influence body size during puberty. Body composition has a genetic component[11–14] and can be influenced by dietary habits[6], social environment, and variation in activity levels[15,16] related to seasons[5]. A recent study reports the effect of a Covid-19 national lockdown on adolescent BMI[57]. We, therefore, recorded the timing of maturation stages for participants in our study, modelled from longitudinal measurements of height and lower-leg length, and variation in these parameters and weight gain related to ancestry, seasons of the year and a Covid-19 lockdown that occurred unexpectedly between the end of March 2020 until the beginning of June when New Zealand returned to Level 1 (https://covid19.govt. nz/assets/resources/tables/COVID-19-Alert-Levels-summary-table.pdf).

Ethical approval for monthly measurements from participants and collection of primary molars was obtained from the University of Otago Human Ethics Committee (approval number H19/030). Research consultation with Māori was obtained from the Ngāi Tahu Research Consultation Committee. In New Zealand, research consultation with Māori is mandated in all areas of research that involve people of Māori descent. Informed consent was obtained from all participants and their parents or guardians. A list of participating schools in Dunedin is given in Acknowledgments.

**Histology**. Thin sections were created following standard procedures[39]. Teeth were embedded in resin (Buehler Epox-iCure®) and sectioned through the tip of the mesial cusp and dentin horn using a Buehler Isomet 1000 precision saw. Sections were fixed to glass microscope slides (Evo Stick® resin), ground (grit P400, P600, P1200) (Buehler® EcoMet 300), polished with a 0.3 μm aluminium oxide powder (Buehler® Micro-Polish II), cleaned in an ultrasonic bath, dehydrated in 95–100% ethanol, cleared (Histoclear®), and mounted with a coverslip (DPX®). Thin section thickness is determined by the visibility of incremental lines. Lines can become visible at different depths in thin sections of a primary molar from different individuals. Sections were examined using a high-resolution microscope (Olympus® BX53) and microscope camera (Olympus® DP25). Images were obtained and analysed in CELL® Live Biology imaging software.

Retzius periodicity data was recorded by GM in the United Kingdom, independently and blind to the New Zealand growth data. Each participant was selected for inclusion in the study if we were able to produce two matching RPs for their primary molars, either: (a) from the outer lateral enamel of each participant's first and second primary molars or (b) from one single primary molar. Lateral enamel commences as the first Retzius line emerges on the outer enamel surface as perikymata (meaning, growth lines on the exterior rather than the interior of the tooth enamel).

**Table 1 Descriptive statistics for RP-biorhythm and growth measures.**

| Participants | n | Retzius periodicity | | Starting age | Maturation stage[a] | | | | Gains | | | Ending | Gain |
|---|---|---|---|---|---|---|---|---|---|---|---|---|---|
| | | mode | mean | yrs | 1. pre | 2. early | 3. peak | 4. late | Weight kg | Height cm | Leg cm | BMI percentile | BMI kg/m² |
| All | 61 | 6 | 7.26 ±1.31 | 10.33 ± 0.57 | 9 | 20 | 27 | 3 | 6.33 ±2.79 | 6.92 ±1.39 | 2.37 ± 0.57 | 69.06 ± 25.44 | 1.13 ±1.04 |
| Female[b] | 34 | 8 | 7.50 ±1.33 | 10.30 ± 0.59 | 1 | 2 | 26 | 3 | 6.97 ±2.82 | 7.39 ±1.55 | 2.41 ±0.47 | 69.19 ±25.64 | 1.31 ±1.06 |
| Male | 27 | 6 | 6.96 ±1.25 | 10.36 ±0.56 | 8 | 18 | 1 | 0 | 5.56 ±2.60 | 6.37 ±0.92 | 2.32 ±0.67 | 68.91 ±25.69 | 0.95 ±1.01 |

[a]Determined from longitudinal measurements of height and weight modelled using fixed bandwidth kernel weighted robust third-degree polynomial regression smoothing.
[b]We were unable to assign a maturation stage to two females.

We found no evidence that RP changed within an individual when compared between their primary molars, either in comparisons between mandibular and maxillary molars or first and second molars (Supplementary Table 1). This is consistent with findings for permanent molars[33]. Oblique thin sections were identified and removed from the study. Oblique sections can be easily identified from the morphology of the dentin horn together with the slope of the enamel buccal and lingual surfaces of the functional and guiding cusps.

RP was calculated in two standard ways. The number of daily cross striations was counted along a prism between two adjacent Retzius lines in lateral enamel at 200-400x magnifications (including the ocular magnification). When consecutive cross striations were not clearly visible between two Retzius lines, RP was calculated from local daily enamel secretion rates (DSRs) divided by prism lengths[45].

We had a good understanding of DSRs in primary molars of these New Zealand children[56]. Variation in DSRs was not a confounding factor in our calculation of RP as DSRs vary only slightly in the outer lateral enamel of primary molars of New Zealand European children[56]. DSRs were calculated by measuring along a prism across the span of six cross striations, which corresponds to 5 days of enamel formation (two adjacent cross striations = 24 h of enamel secretion) and dividing this measurement by five to get a daily mean DSR. This was repeated six times within the local enamel so that a grand mean DSR could be calculated. Following this first calculation, the distance between four to six adjacent Retzius lines was also measured, corresponding to three to five repeat intervals respectively, and divided by three or five. This distance between two adjacent Retzius lines was then divided by the grand mean DSR to yield an RP value.

**Measurements of weight, height, maturation and body mass index**. These were recorded by SW in Dunedin, independently and blind to the Retzius periodicity data that was generated in the United Kingdom. Height, weight and lower-leg length measurements were recorded from each child over a 14-month period between September 2019 to October 2020 during visits to the schools. Most measurements were taken about 4 weeks apart, excluding January 2020 during the school holiday and between March to early June 2020 during the national lockdown due to the onset of the COVID-19 pandemic. Standing height measurements were taken using a Seca 213 Stadiometer. Lower-leg length measurements were recorded three times per participant per visit, using a custom-made laser measuring device with the children in a standardised seated position. Weight was recorded on calibrated scales.

The maturity status of each participant was primarily estimated by modelling longitudinal measurements of their heights taken approximately once per month. Measurements were modelled using fixed bandwidth kernel weighted robust third-degree polynomial regression smoothing of heights on measurement dates[58]. Each individual was assigned one of four maturity scores based upon criteria involving the shape of individually modelled curves along with their sex and age-specific heights. Individuals who were relatively short for age and had not reached pre-spurt minimum height growth velocity were assigned a maturity score of 1 ('pre' in Table 1). Individuals who had reached pre-spurt minimum height growth velocity but who were not near peak height velocity were assigned a maturity score of 2 ('early'). Those individuals who were very close to or who had just exceeded peak height growth velocity were assigned a maturity score of 3 ('peak'). Individuals who had clearly exceeded peak height velocity and were approaching an upper asymptote were assigned a maturity score of 4 ('late'). Individual maturity status was also

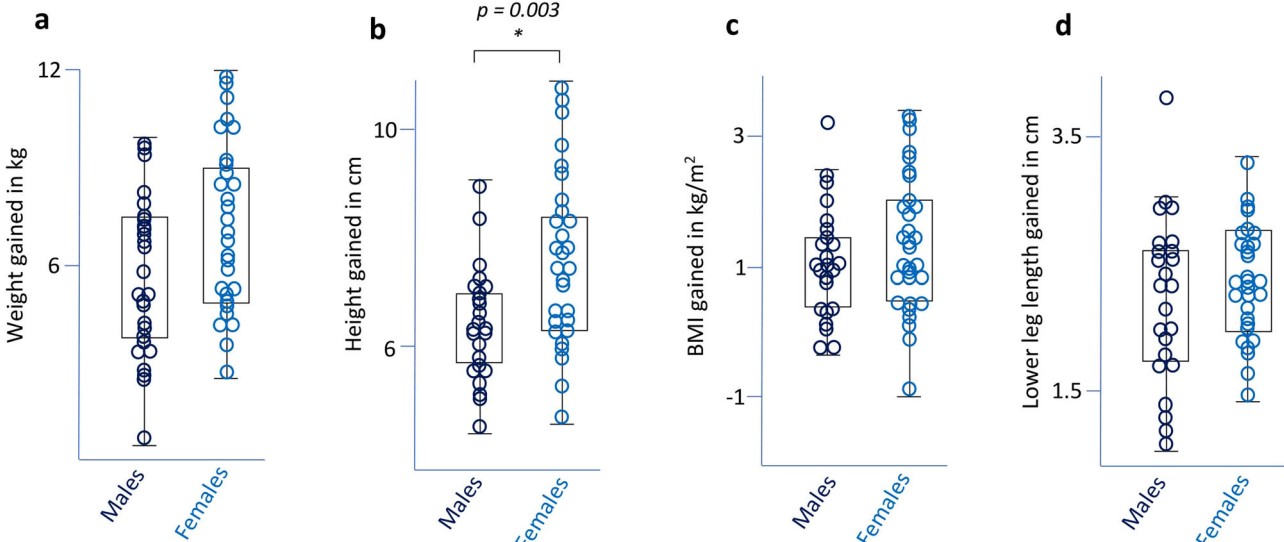

**Fig. 2 Sex differences in growth over 14 months. a** Females ($n = 31$) attained more weight than males ($n = 26$) and the difference approached significance ($p = 0.054$). **b** Females ($n = 31$) gained significantly more height compared to males ($n = 26$). **c** On average, females ($n = 30$) attained a greater but not significant increase in BMI relative to males ($n = 27$). **d** On average, female ($n = 30$) lower-leg length was greater than that of males ($n = 26$). Data are represented as box plots showing interquartile ranges and whiskers that illustrate the minimum and maximum values that were not outliers. *$p < 0.05$, two-tailed $t$-test.

estimated using the same approach but with longitudinal measures of lower-leg length. Results were very similar and are available from the authors.

BMI and BMI percentiles for a given age and sex were calculated using each participant's birth date, sex, height, weight and date that the measurements were taken. These measurements were entered into the online calculator for New Zealand children provided by the New Zealand Ministry of Health.

**Statistical analyses**. Data were log-transformed. Pearson correlation coefficient was used to measure the strength of association between gains in weight gain and height, lower-leg length, starting age and maturation stage. The influence of starting age on the relationship between RP and weight gained over 14 months was assessed through partial correlations. Height and weight were compared between males and females with a two-tailed $t$-test. Weight was compared between females grouped by RP using a Kruskal–Wallis $H$ with multiple comparisons. The relationship between RP and weight/gained over 14 months was modelled using quadratic regression with $p$ values adjusted using a Bonferroni correction. We conducted further analyses using a Kruskal–Wallis $H$-test with multiple comparisons to analyse the rank order of RPs and weight/BMI gained when grouped by those with 6, 7 and 8 days, which were the largest sample sizes. A Chi-square test was used to determine if there was a relationship between participants with RPs of 5 or 6 days and a BMI of or greater than the 95th percentile compared to those with RPs of 7 or 8 days. Multivariate regression was undertaken to assess the relative strength of the effect of log-transformed weight, leg length and stature on the predictor variable RP, using standardised beta coefficients. We also examined the relative relationship of RP to total gains in log-transformed weight, height, and leg length, using standardised beta coefficients, just for females with maturation scores of three.

**Reporting summary**. Further information on research design is available in the Nature Research Reporting Summary linked to this article.

## Results

**Descriptive data**. Participants gained an average of 6.33 kg over 14 months (Table 1). Log-transformed weight gained over this period was positively and significantly correlated with total gains in height ($p = 0.018$) and lower-leg length ($p = 0.006$), but not starting age (in years) ($p = 0.616$) (Supplementary Fig. 1a–c). The average starting BMI of 18.51 kg/m$^2$ (range = 14.9–28.80) is close to the average BMI of 19.8 kg/m$^2$ (range = 14.4–33.8) reported for slightly older children from Dunedin[3]. Within our cohort, weight gained (6.69 kg, sd = 2.82) by New Zealand European females over 14 months (the largest sample size) was similar to weight gained by New Zealand European/Māori and New Zealand European/Pasifika females (6.70 kg, sd = 2.47).

RP-biorhythm had a mean value of 7.26, a modal 6-day periodicity, and a range between 5 to 10 days (Table 1) that lies within the range of RP's reported for humans[37,39,45]. As with permanent molars[33], we found no evidence that RP varied between primary maxillary or mandibular molars or between primary first and second molars when compared within individuals (see Supplementary Table 1). Within the cohort, New Zealand European females had a mean RP of 7.50, that was slightly higher than the mean RP of 7.37 for New Zealand European/Māori/Pasifika females.

**Sex differences in weight, mass and height**. As expected, female weight, BMI, height and lower-leg length increased by a greater amount than males when compared over a 14-month period between September 2019 to October 2020 (Fig. 2a–d and Supplementary Data 1). On average, females weighed more at the start (females = 38.41 kg; males = 37.42 kg) and end of the project (females = 46.00 kg; males = 43.20 kg). Twenty-six females were assigned a maturation score of 3 (Table 1), having probably reached peak height velocity. Eight males were preadolescents, 18 had entered adolescence, and one individual probably approached peak height velocity. As expected, log-transformed maturity scores were significantly and positively correlated with weight/BMI gained over 14 months (Supplementary Fig. 2a-b).

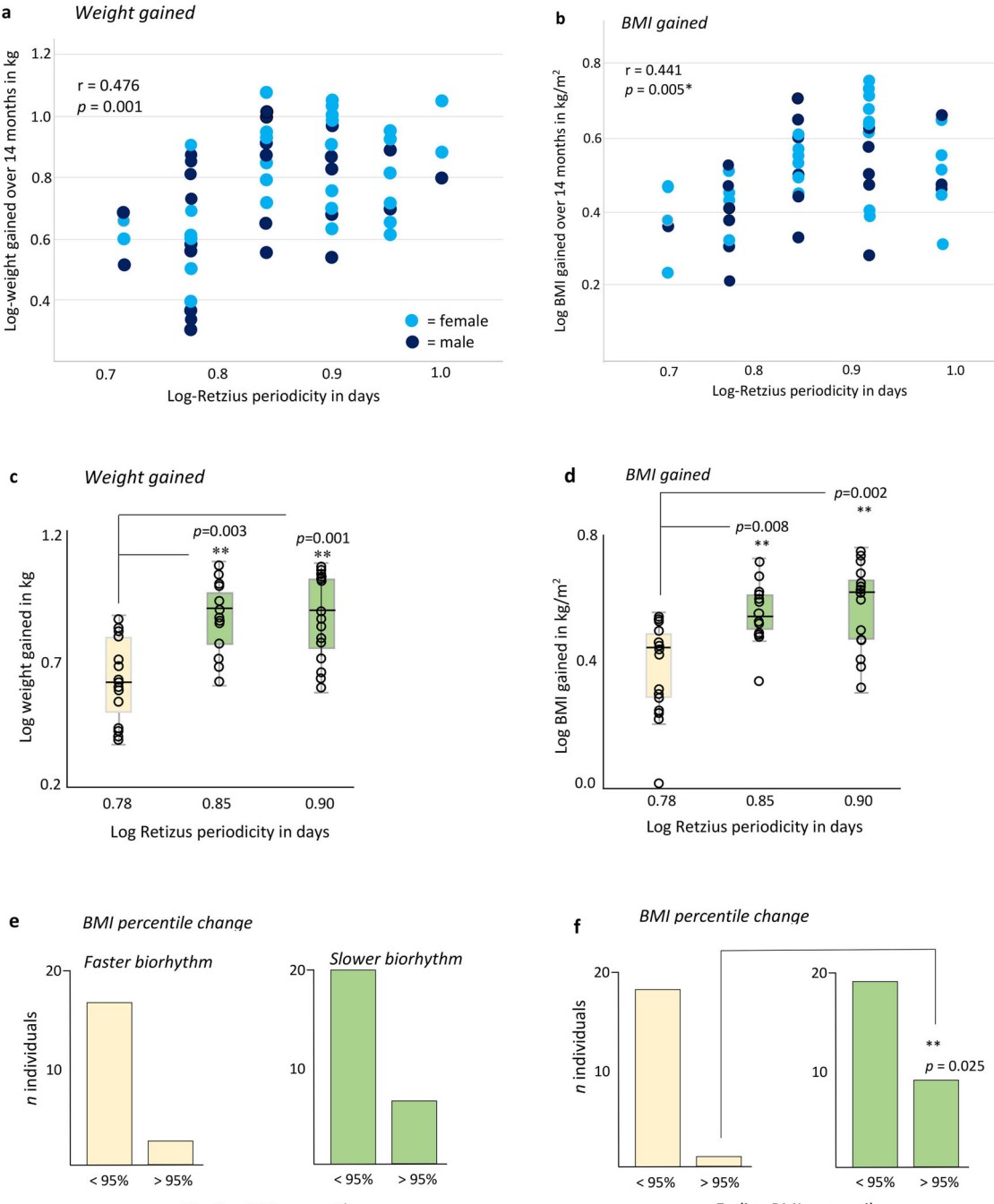

**Fig. 3 Weight and BMI gained relates to RP-biorhythm. a** Scatter plot illustrating that the best way to model the significant relationship between log-transformed weight gained after 14 months and log-RP ($n = 58$) was through a curvilinear quadratic regression model. Excludes outlier. **b** Scatter plot illustrating the significant relationship between log-transformed BMI gained after 14 months and log-RP ($n = 54$). Excludes two outliers and RP of 10 ($n = 2$). **c** Kruskal–Wallis $H$-test with multiple comparisons illustrating the significantly greater gain in weight for those with RP of 7 ($n = 12$) or 8 days ($n = 15$) compared to those with an RP of 6 days ($n = 14$; one outlier removed). **d** Kruskal–Wallis $H$-test with multiple comparisons showing the significantly greater BMI for those with RP of 7 ($n = 12$) or 8 days ($n = 15$) compared to those with an RP of 6 days ($n = 16$; one outlier removed). **e** BMI percentile at the start of the project split into those that have a percentile that is less than 95% and greater than 95% compared to a faster (low RP value) and slower biorhythm (high RP value); and **f** after 14 months, illustrating the significant association between obesity and a slow biorhythm. **\*\***$p < 0.05$. Data are represented as box plots in **c** and **d** showing interquartile ranges and whiskers that illustrate the minimum and maximum values that were not outliers. Bars represent the number of individuals in **e** and **f**.

**Weight and mass gained relate to RP-biorhythm.** Regression analyses revealed log-transformed RP was significantly related to the log-transformed weight and BMI (Fig. 3a, b and Supplementary Data 2) that participants gained over 14 months. A quadratic equation was the best fit for our data as the relationship

between RP and weight/BMI was curvilinear. RP was still significantly related to weight gain over shorter intervals of 12 and 13 months, one longer interval of 15 months, and to adjusted maximum weight gains over 14 months (Table 2). After applying a conservative Bonferroni-corrected criterion to adjust for

**Table 2 Regression analyses of log-transformed gains in weight and body mass index and their association with log-transformed Retzius periodicity.**

| | Quadratic curve | | | | |
|---|---|---|---|---|---|
| RP in days vs: | Intercept | Slope | r | r² | p |
| **Total weight gained in kg** | | | | | |
| Sept 2019 to Aug 2020 | −5.530 | 7.593 | 0.492 | 0.243 | 0.012* |
| to Sep 2020 | −2.614 | 6.796 | 0.498 | 0.248 | 0.007* |
| to Oct 2020ᵃ | −2.876 | 7.526 | 0.476 | 0.227 | 0.001* |
| to Nov 2020 | −3.390 | 8.861 | 0.524 | 0.275 | 0.002* |
| **Total adjusted maximum weight gained in kgᵇ** | | | | | |
| Sept 2019 to Oct 2020ᶜ | −3.126 | 7.849 | 0.483 | 0.233 | 0.000* |
| **Total change in body mass indexᵈ in kg/m²** | | | | | |
| Sept 2019 to Oct 2020 | −2.864 | 7.351 | 0.441 | 0.190 | 0.005* |

ᵃExcludes one extreme outlier.
ᵇLast minus first measurement/time interval.
ᶜExcludes one extreme outlier.
ᵈExcludes one outlier. Variable reflected and then log-transformed. Excludes RP of 10 (n = 2).
*Statistically significant with p < 0.05.

**Table 3 Regression analyses of log-transformed average total weight and associations with log-transformed Retzius periodicity.**

| | Quadratic curve | | | | |
|---|---|---|---|---|---|
| RP in days vs: | Intercept | Slope | r | r² | p |
| **Average weight over 14 months in kg** | | | | | |
| | −0.375 | 4.431 | 0.333 | 0.111 | 0.035* |
| **Average monthly weight in kgᵃ** | | | | | |
| Aug 2020 | −2.132 | 8.829 | 0.401 | 0.161 | 0.026* |
| Sep 2020 | −1.205 | 6.507 | 0.404 | 0.163 | 0.022* |
| Oct 2020 | −1.379 | 6.908 | 0.397 | 0.157 | 0.012* |

ᵃRetzius periodicities of 5 to 9.
*Statistically significant with p < 0.05.

multiple testing, all but one p-value in Table 2 is still significant falling below 0.008 (0.05 divided by six tests). Examination of partial correlations revealed starting age had no influence on the relationship between RP and weight gained (Supplementary Table 2).

We conducted further analyses using a Kruskal–Wallis *H*-test with multiple comparisons to analyse the rank order of RPs of those with 6, 7 and 8-day periodicities (the largest sample sizes) compared to their weight/BMI gained. Participants with an RP of 6 days gained significantly less weight (mean weight = 4.19 kg) after 14 months, compared to the greater average weight gain of those with RPs of 7 days (mean = 7.61 kg) or 8 days (mean = 7.80 kg) (KW = 12.774, df = 2, p = 0.002; Fig. 3c and Supplementary Data 2). Participants with an RP of 6 days also gained significantly less BMI (mean BMI = 0.38 kg/m²) after 14 months, compared to the much greater average BMI gain of those with RPs of 7 days (mean = 1.51 kg/m²) or 8 days (mean = 1.73 kg/m²) (KW = 11.283, df = 2, p = 0.004; Fig. 3d and Supplementary Data 2).

**Mass greater than the 95th percentile relates to RP-biorhythm.** Starting (September 2019) and ending-BMI percentiles (October 2020) from participants with RPs of 5 or 6 days (n = 20) were compared to those with RPs of 7 and 8 days (n = 27). Of the participants with a lower RP, n = 3 had a starting BMI that was equal to or above the 95th percentile compared with n = 7 of those with a higher RP, but the Chi-square test of association was not significant (Fig. 3e and Supplementary Data 2). After 14 months, n = 1 participant with a lower RP had a BMI above the 95th percentile compared to n = 9 of those with a higher RP, and the association was significant (x (1) = 4.755, p = 0.025; Fig. 3f and Supplementary Data 2). Participants with higher RP's were 6.6 times more likely to develop obesity (BMI >95th percentile) after 14 months.

**Average total weight relates to RP-biorhythm.** Regression analyses revealed RP was significantly related to the average weight of the participants over 14 months (Table 3; Fig. 4a and Supplementary Data 3). Examination of the month-by-month average weight of the participants revealed those with repeat intervals of 7 or 8 days typically weighed more each month compared to those with RPs of 6 days (Fig. 4b, c and Supplementary Data 3). Sex differences in weight and RP values are contributing factors here, as n = 17 of those with RPs of 7 and 8 days were females compared to n = 12 males (see analyses below). Further regression analyses revealed RP was significantly related to the monthly weight of the participants (Table 3; Fig. 4d–f and Supplementary Data 3). Applying a conservative Bonferroni correction to adjust for multiple testing, one of the four p values in Table 2 are significant below 0.013 (0.05 divided by four tests).

**Sex differences in RP-biorhythm related to weight gain and total average weight.** Females had a higher modal RP of 6 days compared to the male modal RP of 6 days (Table 1; Fig. 5a; Supplementary Fig. 3 and Supplementary Data 4). Females with a log-transformed RP of 6 days gained significantly less weight over 14 months (KW = 8772, df = 3, p = 0.032; Fig. 5b, c and Supplementary Data 4) and less BMI (KW = 8.829, df = 3, p = 0.032; Fig. 5e) compared to females with higher RPs of 7 to 9 days. For males, the greatest average weight gain occurred with a 7-day periodicity, unlike the greatest average gain for females, that occurred with an 8-day periodicity (Fig. 5b and Supplementary Data 4). Males with RPs of 6 days gained the least weight, but the step-up in periodicity from 6 days did not lead to significantly greater gains in male weight (Fig. 5d and Supplementary Data 4) or BMI (Fig. 5f and Supplementary Data 4), though the relationships were in the expected direction. Thus, the link between RP and weight gain, and BMI gain, is much stronger for females than males, which we interpret here as equivalent to sex differences in the link between enamel formation processes and RP[39].

Females with an RP of 6 days were, on average, lighter over 14 months compared to females with RPs of 7, 8 or 9 days, but the difference was not significant (Fig. 5g and Supplementary Data 4). There was no significant difference in the average weight of males over 14 months when they were grouped and compared by their RP values (Fig. 5h and Supplementary Data 4) though the mean values trended in the expected direction.

We conducted additional analyses to identify the potential effect of covariates on the relationship of RP to gains in weight/BMI.

**RP-biorhythm is related to weight but not height and lower-leg length.** Regression analyses revealed that log-transformed RP was not significantly related to total gains in height (p = 0.225) or lower-leg length (p = 0.165) though the relationships were in the expected direction.

Multivariate regression was undertaken to assess the relative strength of the predictor (RP) on each independent variable (log-

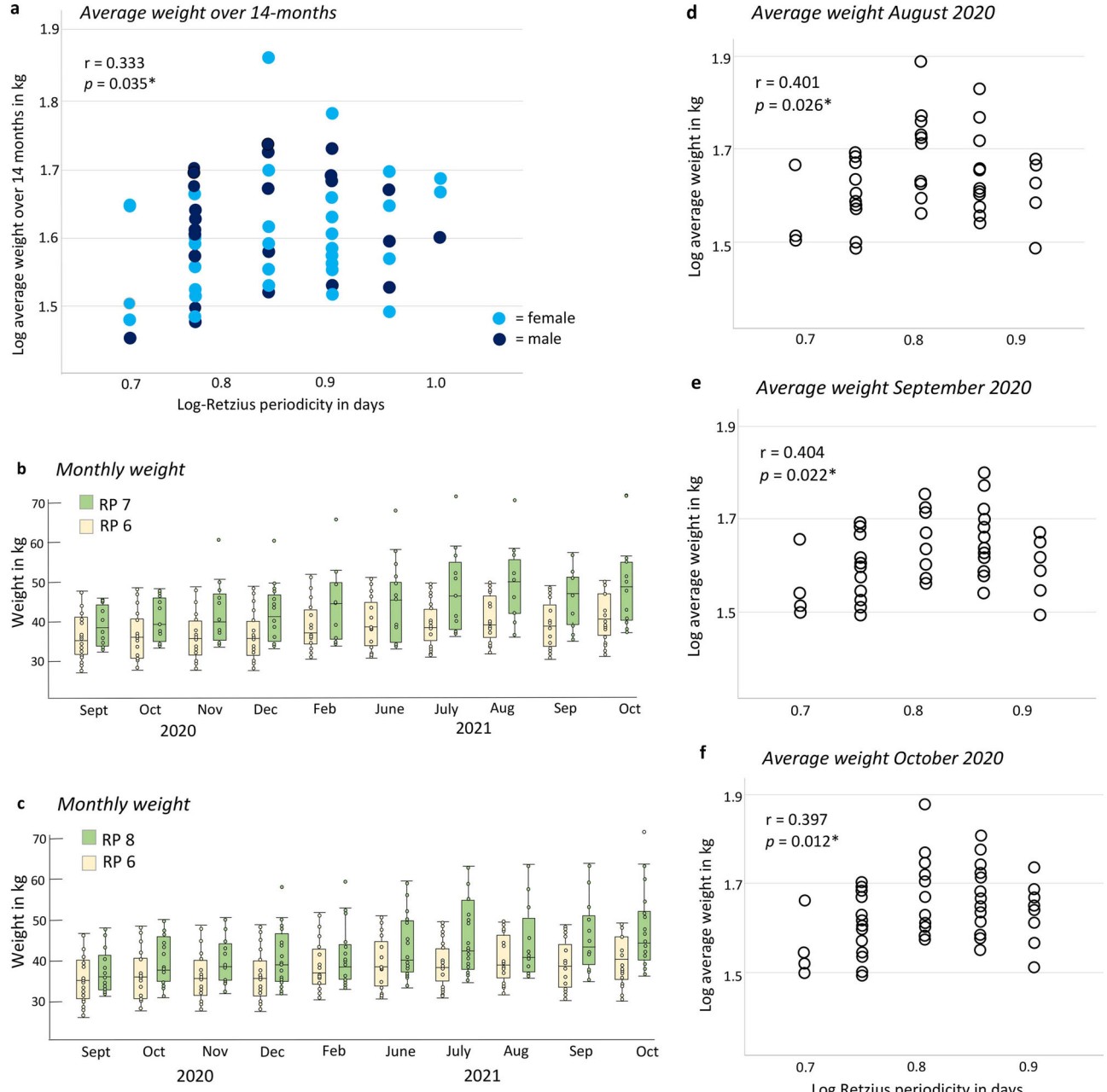

**Fig. 4 Average weight relates to RP-biorhythm. a** Scatter plot illustrating the significant relationship between log-transformed total average weight over 14 months (September 2019 to October 2020) and RP (n = 60) through a curvilinear quadratic regression model (one outlier removed). **b** Monthly weight, and trajectory of weight gain for participants subdivided into those with Retzius periodicities of 6 (n = 17), and 7 days (n = 13); and **c** 6 (n = 17) and 8 days (n = 16). Data were represented as box plots in **b** and **c** show the median value, interquartile range and minimum and maximum values that were not outliers. Quadratic regression models for the average weight of participants with RPs of 5 to 9 in **d** August 2020 (n = 44), **e** September 2020 (n = 46), and **f** October 2020 (n = 55).

transformed gains in weight, leg length and height). Examination of standardised beta coefficients indicated that RP had the strongest effect on weight gained (weight β = 0.315, p = 0.027; height β = 0.005, p = 0.972; lower-leg length β = 0.131, p = 0.400), and only weight gain significantly predicted RP.

**The effect of maturation stage**. Females tended to have a higher RP in this sample and were more mature compared to males. We separated females with a maturity score of three, the largest sample size (n = 26), to determine if the relationship between RP and weight gain persisted after the effect of the maturation stage was held constant. Regression analyses revealed the significant

relationship between log-RP and log-weight gained was still present (Supplementary Fig. 4).

Multivariate regression was undertaken to examine the relative relationship of RP to total gains in log-transformed weight, height and leg length for just these females with maturation scores of three. Examination of standardised beta coefficients indicated that weight had the strongest relationship to RP (weight β = 0.425, height β = −0.178, lower-leg length β = 0.230).

**The effect of Covid-19 lockdown and seasons**. There were substantial differences in the amount of weight gained over lockdown when participants were grouped and compared by

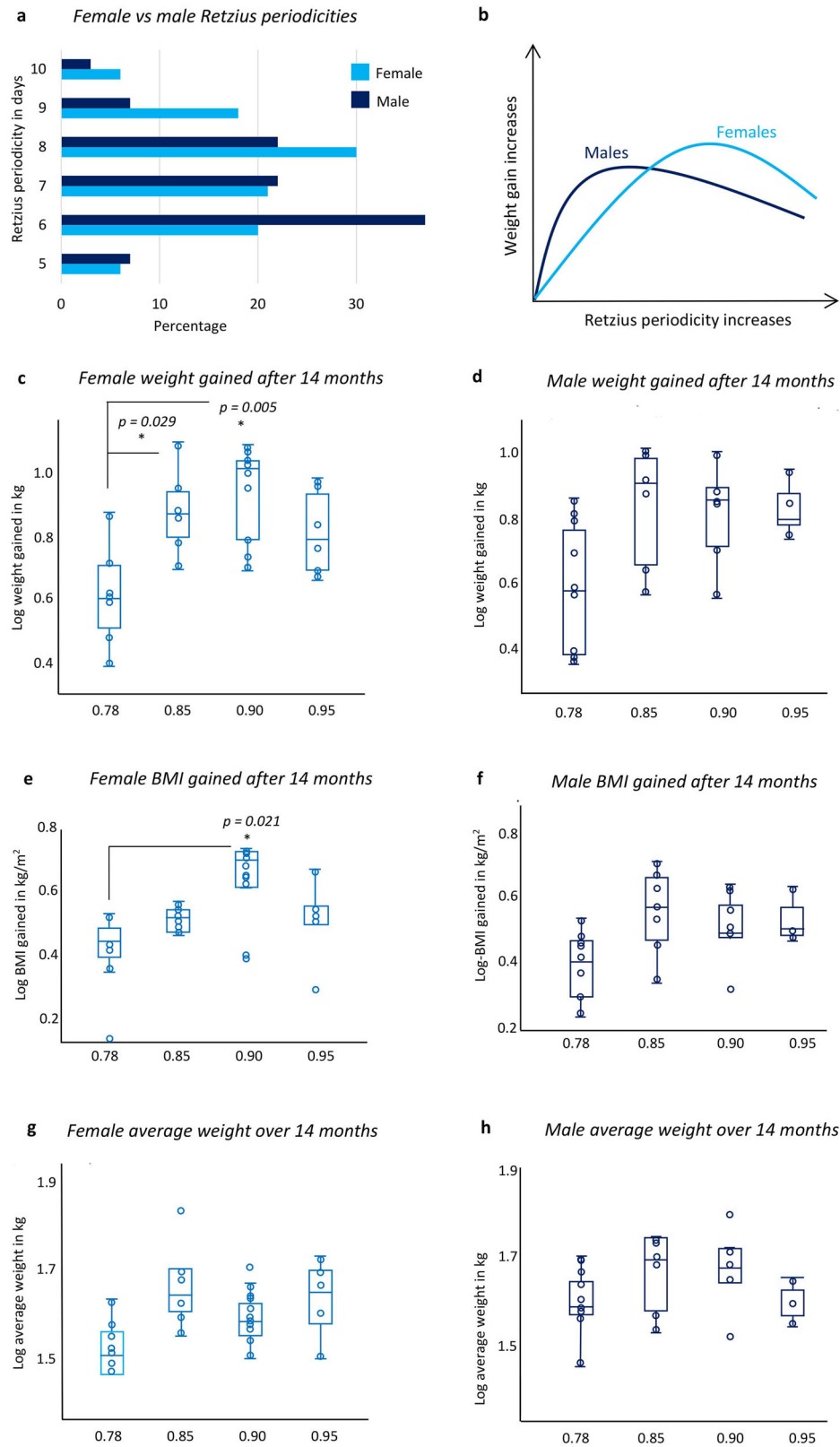

their periodicities (Supplementary Table 3). Those with an RP of 6 maintained a relatively consistent trajectory of weight gain throughout the entire 14-month period (Fig. 4b, c). They gained an average of 1.00 kg during the lockdown, which was similar to the 1 kg gained during the preceding summer period (Supplementary Table 3). However, participants with an RP of 7 gained an average of 3.50 kg during the lockdown, which was at least twice that compared to any other season of the year. Participants with RP's of 8 gained 3.03 kg during the lockdown. These findings suggest pandemic stressors may have been more impactful for adolescents with slower RP-biorhythms.

**Fig. 5 Sex differences in RP-biorhythm related to weight/BMI gain and total average weight. a** Bar chart showing the percentage of male ($n = 27$) and female ($n = 34$) RP values (also see Fig. S3). **b** Line chart illustrating the sex difference in the relationship between RP and weight gained over 14 months. Log-transformed RP values compared to **c** log-transformed female weight gained (female RP of 6 $n = 7$, RP of 7 $n = 6$, RP of 8 $n = 9$, RP of 9 $n = 5$) and **d** male weight gained over 14 months (male RP 6 $n = 9$, RP 7 $n = 6$, RP 8 $n = 5$, RP 9 $n = 3$; one outlier removed). **e** Log-transformed RP compared to log-transformed female BMI gained (female RP 6 $n = 5$, RP 7 $n = 6$, RP 8 $n = 9$, RP 9 $n = 5$) and **f** male BMI gained over 14 months (male RP of 6 $n = 9$, RP of 7 $n = 6$, RP of 8 $n = 5$, RP 9 $n = 3$; one outlier removed). **g** Log-transformed RP values compared to log-transformed female average weight over 14 months (outlier excluded; female RP 6 $n = 6$, RP 7 $n = 6$, RP 8 $n = 10$, RP 9 $n = 5$), and male average weight over 14 months (male RP of 6 $n = 10$, RP 7 $n = 6$, RP 8 $n = 5$, RP 9 $n = 3$). *$p < 0.05$. Data were represented as box plots in **c**, **d**, **e**, **f**, **g**, and **h** showing interquartile ranges, median values and whiskers that illustrate the minimum and maximum values that were not outliers. **c**, **e** Show multiple comparisons.

The summer vacation period coincided with a period of slight weight loss for those with RPs of 7 and 8 days (Fig. 4b, c). On average, those with a 7 and 8-day periodicity gained more weight in the Spring and Winter seasons compared to those with RPs of 6 days (Supplementary Table 3).

## Discussion

Understanding of the relationship between RP-biorhythm and body mass has focused mainly upon interspecific analyses of mammalian species. Although humans retain evidence of this rhythm in hard tissues, the relevance for childhood growth remains underdefined. In this study, we related the timing of the rhythm to mass and weight gained by a cohort of adolescents followed longitudinally. We observed participants with a faster RP-biorhythm (5 or 6-day periodicity) typically weighed less, gained the least weight, and had the smallest change in their BMI compared to those with a slower biorhythm (7 or 8-day periodicity). To our knowledge, our finding provides the first evidence that a long-period biorhythm relates to the rapid change in body size that occurs during puberty.

Our data partly conform with the hypothesised interspecific biological pathway that relates RP to growth[31]. Greater gains in mass are related to higher ('slower') RPs, not the lower ('faster') RPs predicted for humans[30]. Participants with lower RPs gained less weight and mass over the course of the project when comparisons were undertaken between those with periodicities that lay between 5 and 8 days. Limited weight gain suggests a lower RP-biorhythm associated with a less intense growth spurt. Our data differ from the interspecific pathway in that the biorhythm had an optimal periodicity in terms of maximum weight gain during puberty, and this did not relate to the highest RP value. Typically, 7 or 8 day-RPs produced the greatest weight gain. We described this relationship through a curve, not a straight line.

We report an association between a dental biorhythm and weight gain during puberty, but we have not determined how this association relates to the duration of adolescence. Shorter growth periods can combine with more intense growth spurts to allow relatively early maturation[59]. Differences in growth tempo were evident in our sample, but our observations were confined to a 14-month 'window'. It is likely many females entered their growth spurt (pre-spurt minimum velocity) and eventually exited late adolescence at different times, leading to variation in the total duration of their growth periods. Thus, females with 8-day-RPs and more intense growth gained more weight over 14 months, but they might have a shorter adolescent growth period. Conversely, those with a low 6-day RP and a moderate growth rate gained less weight, but they might compensate for this by exiting puberty when they are older. Under this scenario, the biorhythm may have tracked weight via accelerated or decelerated maturation. This would make sense in terms of the reported correlation between RP and adult stature[53–55]. Within the sexes, if females with a 6-day RP-biorhythm mature later, then they should attain greater adult stature[60,61].

We investigated sex differences in the biorhythm as some[39,62] but not all research[63] reports females tend to have higher RP's than males. In compliance with studies that report sex differences in RP, female participants in our study had a higher 8-day modal RP compared to the 6-day modal RP of males (Table 1 and Supplementary Fig. 3). Our finding aligns with expectations for sex differences in the final attained adult human stature[53–55]. It is interesting to note the delayed maturation of males, compared to females, also aligns with their lower modal 6-day RP.

We sought an integrated view of the biorhythm by examining other related measures of growth. While RP was linked to height, as in studies of adult humans[53–55], this link was weak in our data. Peak gains in weight typically follow peaks in height by ~1 year[5]. Many individuals in this study had probably not reached peak weight velocity, but a substantial number of females probably had reached peak height velocity. These relationships might have blunted the influence of RP on height when males and females are considered together.

Environmental influences had modest or temporary effects on our central finding. The weight of many participants decreased over the summer period, but the underlying relationship with the biorhythm returned afterwards. Lockdown led to a period of increased weight gain for those with higher RP's, but the relationship between the biorhythm and weight gain was apparent before, during and after lockdown for these participants.

Excessive weight gain during adolescence can have consequences for adult health[4,64–66]. Excess weight gain during adolescence is more likely to lead to obesity in adulthood[67]. We observed children with higher RPs were six times more likely to be overweight (have a BMI greater than the 95th percentile) after 14 months compared to those with lower RPs. BMI is not a perfect measure of body composition as it can be influenced by body proportions. However, it is related to the percentage of body fat for Dunedin children[3], and BMI is a useful indicator of the way adipose tissue can change during puberty[68,69]. Obesity occurs when energy intake consistently exceeds expenditure[70], which is determined by a complex interaction between genetic and environmental factors[6,13,15,16,71]. It is unsurprising that a hypothalamic-mediated biorhythm is linked to this process. The hypothalamic central melanocortin system responds to hormonal signals from the digestive tract and adipose tissue by regulating food intake and energy expenditure, ultimately impacting body weight[72]. Abnormalities in the melanocortin system, or hormone imbalances, have been linked to early onset human obesity[73,74]. Detailed interrogation of how the RP-biorhythm relates to this system and to genes that are known to associate with obesity and thinness[75,76] should be pursued in future studies.

The biorhythm is related to adolescent weight gain, but the nature of the growing tissue, whether adipose tissue, muscle mass or bone mineral content, has not been established. This is important because body composition during puberty can relate to adult disease[77]. If the type and rate of growth for different tissues correspond with the biorhythm, then new pathways in preventive medicine may be opened, and new approaches developed further

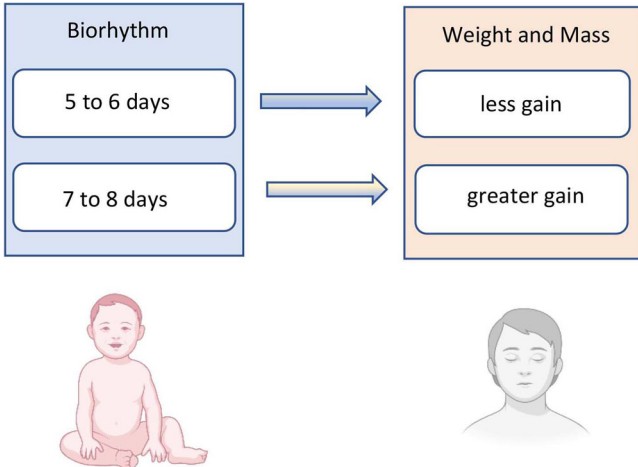

**Fig. 6 Biorhythm in early childhood related to adolescent weight gain.**
Evidence of the biorhythm is captured in primary molars within 2 years of birth, as primary molar enamel forms. A faster biorhythm within 2 years of birth was related to smaller gains in weight and mass during early adolescence. A slower biorhythm is related to greater gains. Part of Fig. 6 (lower panels) was created using a template from BioRender.com (2022).

to explore this long-period rhythm. Studies should also determine if those with histories of early life adversity exposure[78] have fluctuations in RP-biorhythm between early (primary molars) and latter forming teeth (third molars or wisdom teeth) relating to periods of adversity.

The strength of our study lies in the use of direct measures of RP-biorhythm calculated from naturally exfoliated primary molars for each individual, which we compared to measures of the same individual's weight and BMI. A suite of statistical tests allowed us to isolate and identify relationships with the biorhythm and assess the potential effects of covariates on our central finding. Limitations in our study include: (i) growth measurements are descriptive and lack the information and precision of a whole-body scan which would have enabled us to determine which tissue types were responsible for the link between RP and weight gain. (ii) Most male participants were in the early stages of puberty. We could not assess if this determined their weaker associations with RP-biorhythm relative to most females that were at a more advanced stage of puberty. (iii) A practical limitation of our study arose due to the histology methodology. Potentially, $n = 125$ participants were available for the current study (taken from the BCG project), which would have been desirable. However, to ensure an accurate measure of the biorhythm, we required each individual to have two matching RP's, which greatly reduced the sample size. Given that we have now shown RP does not vary between primary molars, as in permanent molars[33], future studies may be able to increase sample sizes by calculating one molar-RP for each individual. (iv) Finally, lockdown increased weight gain for participants with higher RP's, but not for those with lower RPs. It was unclear whether this was behavioural for the higher RP children or an influence of the biorhythm for those with lower RPs. A follow-up study of lockdown behaviour would have helped elucidate this finding.

Our findings raise the possibility that, at least for some individuals, RP-biorhythm may maintain a consistent relationship with aspects of physiology across development (Fig. 6). RP in human primary molars is recorded in enamel within two years following birth[36] and thus reflects processes of development early in life. We observed primary molar RP related to aspects of physical development during early adolescence, which was around 10 years after primary molar enamel had formed. This suggests continuity in the effect of the biorhythm from early life through to adolescence. The sex differences we observed in RP-biorhythm provide further support for this idea, pointing towards biologically based differences that persist across the life course into adulthood or are confined to a given developmental stage where sex differences may be more likely to emerge.

Given the strong association between weight gain and the biorhythm during puberty, it would seem likely that this association could be present during other periods of rapid human growth. Infants gain weight rapidly in the first 6 months after birth. The amount of weight gained during infancy influences the tempo of growth and onset of puberty[79] and is a determinant of obesity in later life[80–83]. The presence of an interspecific association between RP and infant weight[31] points to a biorhythm that might exert an influence on body size from birth. It remains unknown whether this is the case for humans.

Our findings provide researchers with a new avenue from which to explore links between overweight and obese children and adult health risks, as well as an accelerated or decelerated pace of maturation. Naturally exfoliated primary (deciduous or 'milk') teeth from children may prove to be a novel marker of weight-related health risks and thus be an actionable target for intervention many years before adverse health outcomes manifest in adulthood. The aim of developing a novel predictor of human weight and health is clearly worth pursuing.

To summarise, we calculated the timing of a biorhythm in primary molars and compared these values to the weight and mass gained by a cohort of adolescents over 14 months. Those with a faster biorhythm of 5 and 6 days gained the least weight and mass. Those with a slower 7 and 8-day biorhythm were more likely to have a BMI above the 95th percentile. These results provide the first evidence that a long-period biorhythm is associated with adolescent weight gain. Our study points towards a hypothalamic-mediated biorhythm that is active during a key period of human growth.

## Data availability

All data supporting this study and described in this manuscript are available at the University of Kent data repository through the following url: https://data.kent.ac.uk/id/eprint/411.

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

## Acknowledgements

This study was undertaken as part of the Biorhythm of Childhood Growth project funded by The Leverhulme Trust (grant number RPG-2018-226 awarded to P.M.). We acknowledge the European Union's Horizon 2020 research and innovation programme under the Marie Skłodowska-Curie (Grant H2020-MSCA-IF-2018-842812 awarded to A.N.). We thank participants and their families, teachers and principals of the participating Dunedin Schools: Tainui School, Carisbrook School, Tahuna Intermediate, St Bernadette's School, Kavanagh College, George St Normal School, Dunedin North Intermediate, Balmacewen School, Columba College, John McGlashen College, St Claire Primary School, Wakari School, Bathgate Park School, Elmgrove School, Taieri College, Fairfield School, East Taieri School, Arthur St School, and Grant's Braes School.

## Author contributions

P.M., C.L., B.F., and D.G.-S. conceptualised the project. G.M., S.W., C.L., B.F., R.P., and P.M. participated in data generation. P.M. and B.F. conducted data analysis. P.M., E.C.D., A.N., and D.G.-S. wrote the manuscript. All contributed to interpretation.

## Competing interests

The authors declare no competing interests.
