## [Peer Review File · Communications Medicine]

Reviewers' comments:

Reviewer #1 (Remarks to the Author):

The authors present evidence of an association between Retzius periodicity (RP) measured in deciduous molars and weight/BMI gain over a 14 month period during adolescence, within in a moderately sized human cohort based in New Zealand. The manuscript is well written and the methods are sufficiently described for reproduction. Limitations of the study design and possible confounders (e.g. maturity, lockdown) are mostly addressed. Sex differences are explored. I appreciate the breadth of analysis provided in this paper as well as the open access availability of the data and plotting of individual data points. I believe the findings presented will be of interest to the field of anthropology/evolutionary biology and also public health, with a slightly expanded discussion.

Given the sex bias in RP and weight/BMI gain, I would prefer if the sex differentiated results were the primary analysis and presented first, with whole population analysis as a secondary analysis. However, sex differentiated analysis is described in the results and figures, and appropriately discussed in the main paper.

RP was calculated by a single operator (GM) using two methods that are generally accepted as standard. Though some groups may recommend one method over the other, or require measurement by more than one operator to assess error, the aim of this paper is to test the association of RP with weight gain within this population, and therefore I think the methodology is acceptable.

The introduction provides a good overview of biorhythms, pubertal growth and Striae of Retzius. The results presented are in general agreement with the weight of literature reporting a positive interspecific association between RP and body mass across primates and other species. However, there are some intraspecific studies that do not align with this trend that warrant discussion. For example, in a large sample of permanent teeth from Tibet and China, Karaaslan et al. 2021 (PMID: 32657156), found no significant association between RP and adult weight but the relationship tended negative (and RP was negatively associated with height). While this paper does have limitations (tooth type and age do not appear to be accounted for in the analysis), I think it important to include and discuss why the opposite trend may have been observed in this study (e.g. permanent vs deciduous teeth, adolescent vs adult weight, weight gain vs total weight, NZ population vs China/Tibet populations, etc.). The MSc thesis of Simon Chapple (2016 and supervised by PM) also found a negative association between RP and body mass. The hypothesis derived from these findings is that 'faster' RPs should result in more body growth and thus body mass (a negative association also explored in ref #51), and this hypothesis should be discussed in relation to the findings of this paper.

In the present study the authors focus on weight gain, rather than total weight, which is novel and perhaps a strength of the study. Previous studies have used body mass and therefore this paper would benefit from testing the relationship between RP and total weight for comparison against existing literature. Figure 3d/e shows 'slower' RP trended with higher total weight in this study but a supplemental figure showing analysis similar to that provided for weight/BMI gain and sex differentiated, would strengthen the paper.

Deciduous tooth crowns form within the first 2 years after birth and thus RP may provide information about the rate of weight gain during infancy, a factor that has been associated with increased risk of obesity and other risk factors for poor cardiovascular and metabolic health in adulthood. Bromage et al. 2012 (PMID: 22542323) interspecific review found an association between RP and birth weight. I think the paper would be strengthened by a discussion of the

relationship between RP and infant weight/rate of weight gain. While the authors do touch on the health consequences of excessive weight gain during adolescence (lines 447-449), a very brief discussion of literature demonstrating the association of rates of infant weight gain with poor adult health would also be of benefit given this is the period during which the Retzius lines are formed.

Below are minor comments/corrections to address.

I suggest adding number of participants (n) to the different categories in Table 2.

Readability of Figure 3a,b would be improved by colouring data points by sex. I think this is important given the sex bias in RP and weight gained. Figures 3a and S4 would also be improved by using RP in days on the x-axis, rather than log transformed values (as is the case for Fig 3b and Fig 4). The y-axis label for Fig 3b should indicate data is log-transformed.

Fig 4 caption add 'gain' to (b,c,e,f) descriptions (i.e. Log-transformed RP compared to log-transformed male weight gain).

Lines 409-410: Check weight gains over lockdown as values in paper do not match that given in Table S3.

Refs 20 and 21 are duplicated.

Line 100 correct ref 29 to 39.

Refs 58 and 59 are duplicated. The Dunedin ref #58 given on line 198 has been replaced with ref 59.

On line 478, ref 59 is given for the other Dunedin cohort but the reference is to a study in the US.

Please check and correct.

Refs 62 and 64 are duplicated.

Line 458, ref 72 indicates little difference between male and female RPs. I suggest adding other references that support this finding and noting that this is not consistent across all studies (ref 72).

Please check ref 74 on line 479 – a quick skim didn't mention a cohort with 'adversity'.

Ref 75 missing from the manuscript – I suspect it should be added to the methods.

Reviewer #2 (Remarks to the Author):

1, This study has done a good job in hard tissue preparation technology. Due to the specialty, it has a great influence on the observation and measurement of the sample. Please add the thickness range of the slice.

2, In the author's other article "Enamel biorhythms of humans and great apes: the Havers-Halberg Oscillation hypothesis reconsidered" "Applications of the HHO to fossil teeth should avoid transferring RP between deciduous and permanent enamel", but in this study, the source time of RP is the record of the biological rhythm left by the development of the deciduous teeth within 2 years of age. The growth pattern of the body comes from the juveniles. When their deciduous teeth fall off, according to the author's previous point of view, this is the transfer of RP. That is to say, it has been suggested in past research that circadian rhythms that occur during childhood are not suitable for the assessment of later life signs in individuals after growth. contradicts the research in this paper.

3, Human sexual dimorphism, the data from this study have similar trends to what we already know about the HHO hypothesis. In previous experiments, we can see that RP in women are slightly higher than men in the data of adults. "an intraspecific relationship between RP and body size should be negative (Bromage, Juwayeyi, et al. 2016). The duration of growth is variable in interspecific comparisons, while in intraspecific comparisons the duration of growth is less variable. Thus, the intraspecific variations of body size are further under the influence of growth rate than of

the duration of growth. For instance, interspecifically, high RPs indicate growth at reduced rates but for much longer periods of development time to achieve large size, but if the variability in duration of growth is limited, as in this intraspecific study, then growth to larger sizes requires lower RPs that promote an increased rate of cell proliferation.” That is to say, the boy's RP is relatively low, and the rate of promoting cell proliferation is increased. In addition, males mature relatively late, the long bone growth plate closes late, and the adaptive periosteal activity also weakens late, so there will be more deposition, and the large size and body mass of the mature stage is completed. These two points are the key to the difference between men and women, growth rate + time.

In this experiment, the inner relationship is not discussed, only points out that RP is positively correlated with body growth (gains in body weight and body mass), which means that RP is short, body growth less, and a long RP body grows faster. This seems to contradict some research data that observe the life history of adults, which is short RP has high cell activity, more proliferation, and more final construction in a unit time, It is closely related to height. There is a certain logical conflict with the situation observed in this paper. However, these data can also represent a short active period of time, which is itself related to the asynchrony of pubertal developmental rates in males and females, and the difference between males and females is ultimately reflected in the data after the end of their respective maturation.

4. The article is mainly to demonstrate : “A faster biorhythm within two years of birth was related to smaller gains in weight and mass during early adolescence. A slower biorhythm related to greater gains.” However, the observation object of this study is set up in a special biological age stage, puberty and pre-pubertal (entering puberty), which is not suitable for proving the hypothesis. If it is only discussed from the data, the author did demonstrate that there is a special connection between the two. But the cause is unknown. In the discussion, how can there be a credible relationship between RP in childhood and weight gain in adolescence and body mass gain? Is this relationship causal or correlation? need to be further explained.

Logically, there are too many observations of individual differences that correlate with observed RP, such as height, lifespan, cardiovascular disease, bone aging metabolism, etc. That is, if individual RP is obtained, we can simulate and Predict some specific life-history characteristics that are inaccurate. In the HHO theory, RP is a basic internal regulatory mechanism that affects body metabolism and affects body development and growth. Adolescence is a special stage of body development and the most rapid period of body development. The study selected 14 months, to demonstrate that biological rhythms are involved in these aspects of human growth, however, the time period studied was unrepresentative, with girls approaching the peak of maturation, while boys were just entering the second-rapid stage of development. In fact, is it the greater impetus of the RP production, or the lesser contribution of the RP to this result? Body growth and development is not a uniform event, but as the body's endocrine organs and related systems gradually mature, the body is divided into stages, and there are changes in speed. This study selected a life stage with greater changes, and there was a significant difference in developmental speed between boys and girls. Therefore, in this experiment, in order to demonstrate the internal driving force (the driving force of HHO), the selection of observation objects is unreasonable. A clear correlation analysis result was obtained, but this may only be part of the multi-factorial interaction.

The author's team's past research work in recent years has continued to focus on the verification and development of the HHO theory, which is admirable, but this article belongs to a groping and exploratory study, and it cannot rule out other factors to reveal the causal relationship between the two. Observed some findings, but the significance is limited.

Reviewer #3 (Remarks to the Author):

This paper discusses the relationship between the physique of 61 children and RP-biorhythm. The results are clearly presented and the conclusions are hardly controversial.

The following is my personal view, no changes to the manuscript are required.

In this study, body weight and BMI were significantly associated with RP biorhythm, but height was not significantly associated with RP biorhythm. The same results can be seen in other documents. I feel really strange. I wonder if this phenomenon might be due to the effects of thyroid hormones (hyperthyroidism and hypothyroidism).

RESPONSE TO REVIEWERS

Reviewer 1

Overview

The authors present evidence of an association between Retzius periodicity (RP) measured in deciduous molars and weight/BMI gain over a 14 month period during adolescence, within in a moderately sized human cohort based in New Zealand. The manuscript is well written and the methods are sufficiently described for reproduction. Limitations of the study design and possible confounders (e.g. maturity, lockdown) are mostly addressed. Sex differences are explored. I appreciate the breadth of analysis provided in this paper as well as the open access availability of the data and plotting of individual data points.

I believe the findings presented will be of interest to the field of anthropology/evolutionary biology and also public health, with a slightly expanded discussion.

Comments

(1)

Query: Given the sex bias in RP and weight/BMI gain, I would prefer if the sex differentiated results were the primary analysis and presented first, with whole population analysis as a secondary analysis. However, sex differentiated analysis is described in the results and figures, and appropriately discussed in the main paper.

RESPONSE: In the revised manuscript, we have re-structured the Discussion so that males and females are considered separately, rather than grouping them. Males and females are presented separately in the Results.

(2)

Query: RP was calculated by a single operator (GM) using two methods that are generally accepted as standard. Though some groups may recommend one method over the other, or require measurement by more than one operator to assess error, the aim of this paper is to test the association of RP with weight gain within this population, and therefore I think the methodology is acceptable.

RESPONSE: Thank you. And we are confident in our RP values because each individual had to have two matching RP values for inclusion in our study (stated in the methodology).

(3)

Query: The introduction provides a good overview of biorhythms, pubertal growth and Striae of Retzius. The results presented are in general agreement with the weight of literature reporting a positive interspecific association between RP and body mass across primates and other species. However, there are some intraspecific studies that do not align with this trend that warrant discussion. For example, in a large sample of permanent teeth from Tibet and China, Karaaslan et al. 2021 (PMID: 32657156), found no significant association between RP and adult weight but the relationship tended negative (and RP was negatively associated with height). While this paper does have limitations (tooth type and age do not appear to be accounted for in the analysis), I think it important to include and discuss why the opposite trend may have been observed in this study (e.g. permanent vs deciduous teeth, adolescent vs adult weight, weight gain vs total weight, NZ population vs China/Tibet populations, etc.).

RESPONSE: We have followed this reviewer's advice in the revised and extended Discussion.

- We now refer to the study by Karaaslan et a. (2021) on lines 122, 619, and 622.
- We now discuss why the opposite trend may have been observed in our study in new paragraph 3 of the Discussion. We observed an association between a dental biorhythm and weight gain during puberty but we have not determined how this association relates to the duration of adolescence. Our observations were confined to a 14-month window. But the duration of adolescent growth likely varied between the stage 3 females in our sample. Intense growth - which we observed - is linked to shorter growth periods leading to early maturation. So, the females with RPs of 8-days gained most weight, but if they mature early, then their weight gain will stop long before females with 6-day RPs. Thus, the link we observed between weight gain and RP could have been predetermined by variation in growth periods that were observed through a 14-month window. It is possible that the biorhythm is actually tracking weight gain via maturation. Under this scenario, if females with a 6-day RP-biorhythm mature later then they should attain the predicted greater adult stature. This could explain the opposite trend.
- We have split weight gain/growth rates, from total weight in the Discussion and the Results sections.

(4)

Query: The MSc thesis of Simon Chapple (2016 and supervised by PM) also found a negative association between RP and body mass. The hypothesis derived from these findings is that 'faster' RPs should result in more body growth and thus body mass (a negative association also explored in ref #51), and this hypothesis should be discussed in relation to the findings of this paper.

Response: As requested by this reviewer we now outline the hypothesis that a faster biorhythm should result in more human growth and greater mass on lines 121-128 of the Introduction. We then evaluate this hypothesis against our findings in the second and third paragraph of the new discussion.

Simon's MSc thesis (2016) focused on RP and femoral length in medieval adult skeletons not body mass. He inferred adult height from femoral length and sought relationships with RP. The statistically significant negative correlation he reported is now referred to on lines 122, 619, and 625.

(5)

Query: In the present study the authors focus on weight gain, rather than total weight, which is novel and perhaps a strength of the study. Previous studies have used body mass and therefore this paper would benefit from testing the relationship between RP and total weight for comparison against existing literature. Figure 3d/e shows 'slower' RP trended with higher total weight in this study but a supplemental figure showing analysis similar to that provided for weight/BMI gain and sex differentiated, would strengthen the paper.

Response: As requested by this reviewer, in the revised manuscript we have separated weight and BMI gain, from total weight.

- We made a new Table 3 which show RP vs. average total weight over 14 months, and monthly average weight compared to the biorhythm for the months of August, September, October, and November (to match the analyses of weight gain/BMI gain). These are supported by a new set of analyses reported in the Results section on lines 331 to 340. These new analyses support our original findings
- We remade Fig 4, which now focuses on just RP vs. total weight.
- Sex differentiated data is now shown in Fig 3a-3b, Fig 4a, and all of Fig 5.

(6)

Query: Bromage et al. 2012 (PMID: 22542323) interspecific review found an association between RP and birth weight. I think the paper would be strengthened by a discussion of the relationship between RP and infant weight/rate of weight gain.

Response: In the revised manuscript, we have now included a discussion about the relationship of RP and infant weight gain between lines 647-652 and referred to Bromage et al. 2012 on line 651.

(7)

Query: While the authors do touch on the health consequences of excessive weight gain during adolescence (lines 447-449), a very brief discussion of literature demonstrating the

association of rates of infant weight gain with poor adult health would also be of benefit given this is the period during which the Retzius lines are formed.

Response: In the revised manuscript, I now include a brief discussion of the relationship between the amount of weight gained during infancy, which is a determinant of obesity in later life, on lines 695-698.

Minor comments/corrections to address.

(8)

Query: Readability of Figure 3a,b would be improved by colouring data points by sex. I think this is important given the sex bias in RP and weight gained.

Response: I have now added colour to differentiate between the sexes in Fig 3 a, and Fig 3b.

(9)

Query: Figures 3a and S4 would also be improved by using RP in days on the x-axis, rather than log transformed values (as is the case for Fig 3b and Fig 4).

Response: There are constraints with the analysis. I have kept the log transformed values as the analyses was accomplished using log-transformed data which puts the x and y axis onto the same scale. This is not the case with the original Fig 3b or Fig. 4, where we did not use log values.

(10)

Query: The y-axis label for Fig 3b should indicate data is log-transformed.

Response: Done

(11)

Query: Fig 4 caption add 'gain' to (b,c,e,f) descriptions (i.e. Log-transformed RP compared to log-transformed male weight gain).

Response: Done, and this is now Fig 5.

(12)

Query: Lines 409-410: Check weight gains over lockdown as values in paper do not match that given in Table S3.

Response: I have rewritten this section. It is now on lines 576-580.

(13)

Query: Refs 20 and 21 are duplicated.

Response: Done - I have deleted reference 20.

(14)

Query: Line 100 correct ref 29 to 39.

Response: Done - I have changed the reference to 39.

(15)

Query: Refs 58 and 59 are duplicated.

Response: Done - I have deleted reference 59.

(16)

Query: The Dunedin ref #58 given on line 198 has been replaced with ref 59. On line 478, ref 59 is given for the other Dunedin cohort but the reference is to a study in the US. Please check and correct.

Response: I have changed reference 58 to Taylor et al., 2003.

(17)

Query: Refs 62 and 64 are duplicated.

Response: I have deleted reference 64

(18)

Query: Line 458, ref 72 indicates little difference between male and female RPs. I suggest adding other references that support this finding and noting that this is not consistent across all studies (ref 72).

Response: Done – I have noted this is not consistent across all studies and added Tan et al. 2017, on line 575.

(19)

Query: Please check ref 74 on line 479 – a quick skim didn't mention a cohort with 'adversity'.

Response: Done – That should have been reference 73. I have changed this.

(20)

Query: Ref 75 missing from the manuscript – I suspect it should be added to the methods.

Response: Done - I have deleted this reference.

Reviewer 2

(1)

Query: This study has done a good job in hard tissue preparation technology. Due to the specialty, it has a great influence on the observation and measurement of the sample. Please add the thickness range of the slice.

Response: Thank you. We have added to the methodology that thin section thickness is determined by the visibility of incremental markings not a pre-set thickness.

(2)

Query: In the author's other article "Enamel biorhythms of humans and great apes: the Havers-Halberg Oscillation hypothesis reconsidered" "Applications of the HHO to fossil teeth should avoid transferring RP between deciduous and permanent enamel", but in this study, the source time of RP is the record of the biological rhythm left by the development of the deciduous teeth within 2 years of age. The growth pattern of the body comes from the juveniles. When their deciduous teeth fall off, according to the author's previous point of view, this is the transfer of RP. That is to say, it has been suggested in past research that circadian rhythms that occur during childhood are not suitable for the assessment of later life *signs in individuals after growth*. contradicts the research in this paper.

Response:

Assessment of individuals after growth was not an aim in the present study.

Furthermore, we disagree with this Reviewers interpretation of the manuscript by Mahoney et al 2017 in Journal of Anatomy. We did not suggest that RP of deciduous teeth are not suitable for assessment of later life. The statement that "studies should avoid transferring RP between deciduous and permanent enamel" is a methodological recommendation.

- We have showed in Mahoney et al (2017, J Anat) that deciduous enamel of humans and great apes can have a lower RP than permanent enamel.
- We have shown in McFarlane et al. (2021, Am J Phys Anth) that human molar permanent RP can also differ when compared to incisor RP.

These studies have a methodological implication that means you analyse the different tooth types separately. We standardised our methodology in the present study to deciduous molars, and specifically showed that RP did not vary between these molars (Supplementary Table S1). The meaning behind the variation in RP between some deciduous and permanent teeth, and between different tooth types for some individuals but not others, is not known. It was not a goal of the present study to assess why this occurs.

(3)

Query: Human sexual dimorphism, the data from this study have similar trends to what we already know about the HHO hypothesis. In previous experiments, we can see that RP in women are slightly higher than men in the data of adults. “an intraspecific relationship between RP and body size should be negative (Bromage, Juwayeyi, et al. 2016). The duration of growth is variable in interspecific comparisons, while in intraspecific comparisons the duration of growth is less variable. Thus, the intraspecific variations of body size are further under the influence of growth rate than of the duration of growth. For instance, interspecifically, high RPs indicate growth at reduced rates but for much longer periods of development time to achieve large size, but if the variability in duration of growth is limited, as in this intraspecific study, then growth to larger sizes requires lower RPs that promote an increased rate of cell proliferation.” That is to say, the boy's RP is relatively low, and the rate of promoting cell proliferation is increased. In addition, males mature relatively late, the long bone growth plate closes late, and the adaptive periosteal activity also weakens late, so there will be more deposition, and the large size and body mass of the mature stage is completed. These two points are the key to the difference between men and women, growth rate + time. In this experiment, the inner relationship is not discussed, only points out that RP is positively correlated with body growth (gains in body weight and body mass), which means that RP is short, body growth less, and a long RP body grows faster. This seems to contradict some research data that observe the life history of adults, which is short RP has high cell activity, more proliferation, and more final construction in a unit time, It is closely related to height. There is a certain logical conflict with the situation observed in this paper. However, these data can also represent a short active period of time, which is itself related to the asynchrony of pubertal developmental rates in males and females, and the difference between males and females is ultimately reflected in the data after the end of their respective maturation.

Response: The reviewer refers to a hypothesis by Bromage et al 2009 (Calcif Tiss Int) that:

- **was created on adult data.**
- **had not been tested on human adolescent weight gain until our current study.**

In the revised manuscript we outline the hypothesis on lines 121-126 in the Introduction. We

then evaluate it against our data in the second paragraph of the new Discussion.

- The relationship of RP to female weight gain and average weight does not support the hypothesised outcome for humans. A faster biorhythm is hypothesised to result in more body growth and thus greater body mass for humans. We found the opposite. Our data aligns with the interspecific hypothesis which is discussed in paragraph 3 of the new Discussion.

In the revised discussion in paragraph 3 of the Discussion we consider a reason why our data for adolescents might differ to the hypothesis created on adult humans

- We observed an association between a dental biorhythm and weight gain during puberty but we have not determined how this association relates to the duration of adolescence. Our observations were confined to a 14-month window. But the duration of adolescent growth likely varied between the stage 3 females in our sample. Intense growth - which we observed - is linked to shorter growth periods leading to early maturation. So, the females with RPs of 8-days gained most weight, but if they mature early, then their weight gain will stop long before females with 6-day RPs. Thus, the link we observed between weight gain and RP could have been predetermined by variation in growth periods that were observed through a 14-month window. It is possible that the biorhythm is actually tracking weight gain via maturation. Under this scenario, if females with a 6-day RP-biorhythm mature later then they should attain the predicted greater adult stature. This could explain the opposite trend.

(4)

Query: The article is mainly to demonstrate : “A faster biorhythm within two years of birth was related to smaller gains in weight and mass during early adolescence. A slower biorhythm related to greater gains.” However, the observation object of this study is set up in a special biological age stage, puberty and pre-pubertal (entering puberty), which is not suitable for proving the hypothesis.

Response: The reviewer refers to our Fig. 5 which is a descriptive statement based upon our results that has hypothetical implications for future research. This was not an initial hypothesis that we tested (and see response to Query 5).

(5)

Query: If it is only discussed from the data, the author did demonstrate that there is a special connection between the two. But the cause is unknown. In the discussion, how can there be a credible relationship between RP in childhood and weight gain in adolescence and body mass gain? Is this relationship causal or correlation? need to be further explained.

Response: In the revised manuscript (on page 17 paragraph 2, and page 18 paragraph 1 and 2) we have written a new section that considers biorhythm as a biomarker that persists through the life course.

- We observed RP-biorhythm related to aspects of physical development during early adolescence.
- We observed lower RPs of males compared to females was consistent with sex differences in adult size.

- We observed in a previous study that RP related to the size of microscopic canals that house blood vessels in age-matched adolescent ribs (Pitfield et al., 2018).
- Three other studies observed permanent molar RP related to adult stature which strongly suggests the biorhythm tracks adult body size (Chapple, 2016, Bromage et al., 2016, Karaaslan et al., 2021).

All of these studies suggest the biorhythm may exert an influence throughout the life course.

References referred to in response to Query 5.

Pitfield, R., Miskiewicz, J.J. & Mahoney, P. Microscopic markers of an infradian biorhythm in human juvenile ribs. *Bone*, **120**, 403–410 (2018).

Chapple, Simon (2016) *Long period growth lines in enamel and body size in humans: a test of the Havers-Halberg Hypothesis*. Master of Science by Research (MScRes) thesis, University of Kent

Bromage TG, Juwayeyic YM, Katrisa JA, et al. (2016) The scaling of human osteocyte lacuna density with body size and metabolism. *C R Palevol* 15, 32–39.

Hakan Karaaslan, Jeffrey Seckinger, Amel Almagbrok, Bin Hu, Hui Dong, Dengsheng Xia, Tsering Dekyi, Russell T. Hogg, Jian Zhou & Timothy G. Bromage. 2021. Enamel multidienn biological timing and body size variability among individuals of Chinese Han and Tibetan origins. *An Hum Biol.* 48: 23–29

(6)

Query: Logically, there are too many observations of individual differences that correlate with observed RP, such as height, lifespan, cardiovascular disease, bone aging metabolism, etc. That is, if individual RP is obtained, we can simulate and Predict some specific life-history characteristics that are inaccurate.

Response: No study has reported links between the biorhythm and human cardiovascular disease, life span or bone aging metabolism. Human RP correlates with adult height, and now adolescent weight.

(7)

Query: Adolescence is a special stage of body development and the most rapid period of body development. The study selected 14 months, to demonstrate that biological rhythms are involved in these aspects of human growth, however, the time period studied was unrepresentative, with girls approaching the peak of maturation, while boys were just entering the second-rapid stage of development. In fact, is it the greater impetus of the RP production, or the lesser contribution of the RP to this result? Body growth and development is not a uniform event, but as the body's endocrine organs and related

systems gradually mature, the body is divided into stages, and there are changes in speed. This study selected a life stage with greater changes, and there was a significant difference in developmental speed between boys and girls. Therefore, in this experiment, in order to demonstrate the internal driving force (the driving force of HHO), the selection of observation objects is unreasonable. A clear correlation analysis result was obtained, but this may only be part of the multi-factorial interaction.

Response: We agree with the reviewer that a period of 14-months did not capture the total duration of adolescence. **In the revised manuscript we have addressed this point in paragraph 3 of the Discussion.**

(8)

Query: The author's team's past research work in recent years has continued to focus on the verification and development of the HHO theory, which is admirable, but this article belongs to a groping and exploratory study, and it cannot rule out other factors to reveal the causal relationship between the two. Observed some findings, but the significance is limited.

Response: We believe our study is significant because we have observed a new influence on adolescent growth.

Reviewer 3

Overview

This paper discusses the relationship between the physique of 61 children and RP-biorhythm. The results are clearly presented and the conclusions are hardly controversial.

The following is my personal view, no changes to the manuscript are required.

Comments

Query: In this study, body weight and BMI were significantly associated with RP biorhythm, but height was not significantly associated with RP biorhythm. The same results can be seen in other documents. I feel really strange. I wonder if this phenomenon might be due to the effects of thyroid hormones (hyperthyroidism and hypothyroidism).

Response: We suspect the strong relationship between the biorhythm and weight/BMI occurs because the biorhythm is tracking adipose tissue. We are pursuing this proposal in a follow-on study.

REVIEWERS' COMMENTS:

Reviewer #1 (Remarks to the Author):

The authors have satisfactorily addressed my comments. The additions to the introduction and discussion clarified their hypothesis and the seeming discrepancies with other literature. I recommend the revised manuscript be accepted.

Reviewer #2 (Remarks to the Author):

After revising, the analysis of the research data is more reasonable, and, in the conclusion, the author simplifies the meaning of this study and indicates a reasonable direction of the meaning of, "evidence that along-period biorhythm associates with adolescent weight gain " discovery of an interesting phenomenon on the basis of the study of the individual growth and development of the organism itself, and the statistics work very well. However, this report does not obviously change our understanding of the mechanisms of human body growth and development. In other words, the mechanism behind this discovery has not been reasonably revealed and elucidated.

I still think this study has doubts in the selection of the study objects, most of them in adolescence and (early puberty in boys) and a very short time frame of observation, the article is also very clear about this, especially shown in the "Strengths and limitations" section, Such selection of subjects does not lead to the conclusions reasonably expected by the authors.

It is a good vision to propose a new development a novel predictor of human weight and health to make earlier predictions on the risk of metabolic diseases, such as obesity, cardiovascular disease, diabetes, etc., that accumulate in adolescence. In fact, it is relatively in terms of risk prediction of these genetically predisposed diseases, the clinic usually prefers to directly investigate the health history background of children's family members and genetic screening.